# A Preliminary Study of 3D Vernacular Documentation for Conservation and Evaluation: A Case Study in Keraton Kasepuhan Cirebon

**Shafarina Wahyu Trisyanti [1,\*], Deni Suwardhi [1], Iwan Purnama [2] and Ketut Wikantika [1]**

[1] Department of Geodesy and Geomatics, Faculty of Earth Science and Technology, Institut Teknologi Bandung, Jl. Ganesha No.10, Bandung 40132, Indonesia

[2] Department of Architecture, Sekolah Tinggi Teknik Cirebon, Cirebon 45131, Indonesia

\* Correspondence: shafarina.wahyu@students.itb.ac.id

**Abstract:** Despite the wealth of cultural heritage objects in Indonesia, some of which are considered UNESCO World Heritage sites, more documentation still needs to be done. One of the reasons behind this problem is that the documentation of complex cultural heritage objects is more complicated than most modern objects, which are often more simplistic. This preliminary study aimed to document vernacular heritage buildings in 3D to be used as a conservation and building management tool. The built digital model can then be used as a building assessment tool. The data acquisition method used in this study was a combination of photogrammetry and laser scanner technology. The building model was stored as BIM (IFC model) and then georeferenced before being converted to IndoorGML. The building elements' information contained in the building model was retrieved as input to analyze the building. This research included analyses for building assessment, natural room temperature, natural lighting, and indoor space and relation. All results from the analysis were used as input to calculate the reliability value of the building using the AHP method. The case study for the heritage building was the house of Prince Arya Denda at Kasepuhan Palace, Cirebon, Indonesia.

**Keywords:** 3D documentation; building assessment; building model; BIM; IndoorGML; vernacular

## 1. Introduction

Indonesia has a variety of cultural heritage objects, with some registered as world cultural heritage objects by UNESCO. These cultural objects can be in the form of vernacular areas, heritage buildings, or other historical vernacular objects. A vernacular building is defined as the traditional and natural way communities house themselves. It is a continuous process, including necessary changes and adaptation in response to social and environmental constraints [1]. The architectural style of vernacular and modern buildings has many differences, whereas vernacular buildings have an identity in their time and carry elements of local culture. Most vernacular buildings have a more complex architectural form than modern ones; so, they have a higher level of difficulty in documenting them. This study aimed to document vernacular heritage buildings in a 3D model to be used as a conservation and building management tool. The building model can be used for building assessment tool; in this study, we analyzed natural room temperature, natural lighting, and indoor space and relation.

The heritage building used as a case study in this research was a building that currently functions as a residence. The building is in a dense residential area of Kasepuhan Palace, as seen in Figure 1. Kasepuhan Palace is located in Cirebon City, West Java, Indonesia. Kasepuhan Palace, referred to as Keraton Kasepuhan, was established in 1529 and is a cultural heritage center with an important role in history. Keraton Kasepuhan still exists similarly to other palaces in Indonesia. The settlements around the Keraton Kasepuhan were initially formed for families and courtiers [2]. The building that was the case study

in the Keraton Kasepuhan area is a residence belonging to the family of Pangeran Arya Denda, one of the princes. This building is estimated to have been built in 1898 in the village of Mandalangen. According to Said (2004), traditional houses were built in the same way by previous residents without any changes; so, traditional houses were formed based on the traditions that existed in the community. Traditional homes are also referred to as traditional houses, original houses, or people's houses [3].

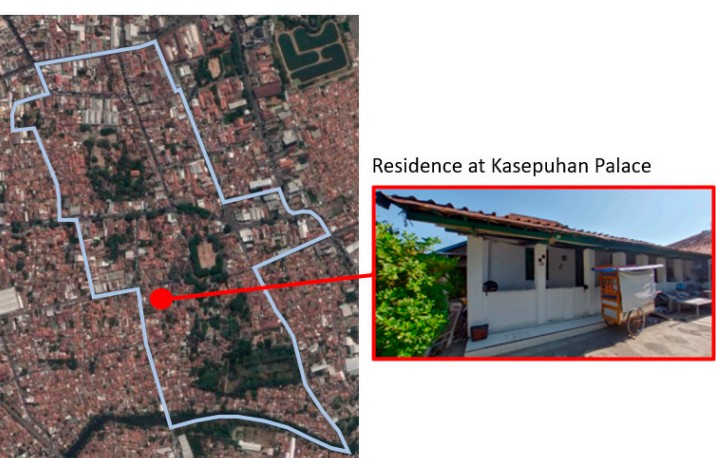

**Figure 1.** Heritage buildings as case studies.

## 2. Literature Review

### 2.1. Digital Representations of Spatial Data

Objects in the real world can be represented or modeled using 3D geographic information system (3D GIS) and building information modeling (BIM) technology. BIM is a digital representation that shares knowledge resources for information about a facility, forming a reliable basis for decisions during its life cycle [4]. BIM working on a 3D model contains information via an intelligent database and can restore geometric and semantic information [5]. Industry foundation classes (IFC) is a standard data storage format developed by BuildingSMART in BIM, which defines the data schema and the structure of the file exchange format [6]. GIS is an information system with specific capabilities for working with spatially referenced data [4] and can be used to model larger areas than BIM [7]. The 3D GIS is an ideal tool for representing 3D geometry, semantics, and topology [5]. One of the data models that is often used in 3D city mapping in GIS is CityGML. CityGML is a data model with Extensible Markup Language (XML) for storage and exchange on a virtual 3D city model [8]. Currently, BIM and GIS are considered separate technologies and challenging to integrate, even though they are necessary to model cities [7]. The 3D model in BIM has rich information that can support building facility management but it works on local coordinates. On the other side, a 3D model in GIS with geographic coordinates can be used for spatial analysis of the larger area. The integration of those technologies can be beneficial for urban planning and evaluation.

Heritage building information modeling (HBIM) is implementing the BIM concept for heritage buildings, whereby interactive parametric data representing architectural elements are integrated and accurately mapped onto 3D scan data. Similar to BIM, HBIM contains various data and information regarding heritage objects from multiple domains of expertise. Being a collaborative platform, HBIM enables architects, archaeologists, art historians, restorers, photographers, and other experts to access the same integrated pool of information [9]. HBIM has the general advantages of BIM, such as the possibility to perform prototyping, visualization, collaboration, energy simulation, comparison of different design options, solar study, and energy demand prediction. However, HBIM is tailored explicitly for heritage buildings, thus providing automated documentation in engineering drawings for precise conservation. It also provides a review of the building's exterior and interior. All these advantages lead to the ease and optimization of the management, survey renovations,

and eventual changes to the structure, rendering these processes much more effective and efficient [10].

For indoor standard spatial data models, there is IndoorGML published by OGC. IndoorGML is a standard format for storing interior building data based on XML, which can provide spatial information to determine the position of objects in buildings and their use in various aspects [11]. IndoorGML defines an indoor space model rather than a feature model for indoor spaces; so, it can be easily extended to meet the requirements of indoor spatial applications and integrated with another standard, such as CityGML [12]. In this research, IndoorGML can be used for indoor spatial analysis.

*2.2. Related Works*

There have been many studies on the 3D modeling of buildings, including heritage buildings. Some studies not only build 3D models but also manage databases and other applications. Basir et al. [13] developed a GIS database for heritage building conservation that contains spatial and attribute data from a 3D model. The authors used a terrestrial laser scanner (TLS) for building the exterior and interior. The 3D modeling process in this study was performed manually based on the size of the registered point cloud. The resulting 3D model had a detailed structure and shape and was stored as the GIS model for conservation purposes.

The laser scanner technique was also used for cultural heritage [14]. In Murphy et al. [15], construction parametric objects/components of heritage buildings based on 18th-century architectural manuscripts were carried out. These objects were stored as GDL objects and then collected in the library for HBIM. The object was then mapped or correlated to the point cloud from the laser scanner data so that a 3D model of the building could be obtained and used as an architectural document.

In addition to using a laser scanner, photogrammetry is another data acquisition technique that can be used for building modeling. Photogrammetry enables many tools to give virtual casts of reality by showing it in the way of the point cloud. In Pulcrano et al. [16], the camera system was used to document the Church of Rosario di Palazzo. The system can be an alternative for heritage documentation, placed at an intermediate level between range-based and image-based traditional technologies. Close-range photogrammetry (CRP) also contributes to measured drawing, reconstruction, and restoration projects for historical buildings in a 3D model [17].

Some study areas use a combination of laser scanning and photogrammetry. The benefit of integrating these two technologies is to take advantage of the terrestrial laser scanner (TLS) capability to directly acquire dense colored clouds with the flexibility of the photogrammetry to operate even in exceptional conditions [18]. In Fawzy [19], the combination of TLS and CRP showed the optimum achieved accuracy for 3D digital documentation and it is recommended for surveying instead of the traditional method. The model of a cultural heritage building from laser scanners and photogrammetry can be visualized at a different level of detail using heritage building information modeling (HBIM) for building maintenance [20].

Building reliability can be evaluated manually using the weighting method [21]. The data used are primary data obtained by performing the measurement and visual documentation of the object of study as well as secondary data such as documents of working drawings. Building evaluation is carried out by establishing building reliability criteria according to regulations and assigning an assessment weight. The aspects assessed are architecture, structure, utility, accessibility, building, and environmental planning. The disadvantage of this evaluation process is that the appraiser must visit the object directly in his or her assessment. However, the weighting method used can facilitate the analysis of building reliability.

The use of IndoorGML in evaluating the reliability of buildings in [22] was carried out to analyze the accessibility in the building. The building model available in IFC is converted into IndoorGML so that it can be displayed in the form of a node-relation graph.

The graph is then used to find the distance from one room to another, which will be one of the parameters in evaluating the reliability of the building.

In Suwardhi et al. [23], a 3D city map for documenting and managing cities for cultural heritage was created using a combination of photogrammetry and laser scanning and involved the community in participatory mapping. The 3D city map can be used for urban spatial analysis and building spatial analysis.

## 3. Materials and Methods

### 3.1. Digital Models

The city mapping for the area of the palace in Cirebon was performed in previous research, Heritage Smart City Planning (HESTYA) [23], and generated LOD1 and LOD2 of a 3D city model. The acquisition method for city mapping uses a combination of UAV photogrammetry and laser scanner technology. Laser scanning technology is the most often used [24] as it is suitable for indoor and outdoor acquisitions. However, other techniques such as photogrammetry, both by traditional close-range terrestrial cameras and by UAVs (unmanned aerial vehicles) or drones, are also widely used [25] to complement laser data. These techniques are complementary and their integration may present a valuable method for the 3D documentation of complex vernacular heritage with smaller complexes within the aerial mapping of the city, which can produce orthophoto and digital elevation model (DEM) [26].

The acquisition method for building mapping uses a terrestrial laser scanner (TLS). The generated point cloud from the laser scanner can then be used for modeling the building. The building model that has been made can help in making the as-built drawing of the building. Currently, there are many buildings in Indonesia, especially heritage buildings, that do not yet have as-built drawings. For the case study of the house of Pangeran Arya Denda, there was already an architectural drawing that was built using measurements from a distometer and measuring tape. However, the shape of the roof from this architectural drawing was only an estimate; so, a laser scanner model was needed to complete it. The architectural drawing of the residence of Pangeran Arya Denda, by manual measurement, can be seen in Figure 2. Both data types can be combined to produce detailed 3D models of exterior and interior buildings. The architectural drawing can be used as the reference for a geometric validation of a generated 3D model.

The building model in this paper was stored as IFC and IndoorGML models. The IFC model still uses local coordinates; so, it was necessary to georeference it before converting it to an IndoorGML data structure. An IndoorGML model was needed to perform spatial analysis indoors, which cannot be performed using the IFC model. The flow diagram for 3D modeling, starting from data acquisition, can be seen in Figure 3. The 3D modeling used ArchiCAD (Graphisoft) software, and both georeferencing and converting the model from IFC to CityGML used FME Workbench from Safe Software.

Residential buildings in the Kasepuhan residential area are influenced by Sundanese, Javanese, and colonial (Dutch) architectural styles. The identification of the cultural heritage buildings can be seen from the floor plan, foot shape, body shape, head shape, and home decoration [3]. Symmetrical house plans interpret colonial architectural plans. The foot of the house, which is seen using a pedestal foundation with river stone, is found on the southern terrace and pavilion. This foundation is often found in traditional Javanese and Sundanese buildings. On the body of the building, there are large columns and thick walls as a characteristic of colonial architecture; but, in the pavilion, there are eight wooden columns similar to traditional Javanese architecture. In addition to the columns, there are other characteristics of colonial architecture, such as large openings. As for the head of the building, there are three types of roofs, namely, Buka Palayu (Sundanese architecture), Limasan on the terrace (Javanese architecture), and adaptation of the pyramid shape with the addition of Gevel (colonial architecture). The decorations found in this building are similar to traditional Javanese and Sundanese houses (wayang calligraphy, diamonds, etc.) [3]. Green painted doors and windows are the hallmarks of the original Cirebon

dwelling. This residence has a high roof so that the air in the room is cooler. Some of the traditional building parts can be seen in Figure 4.

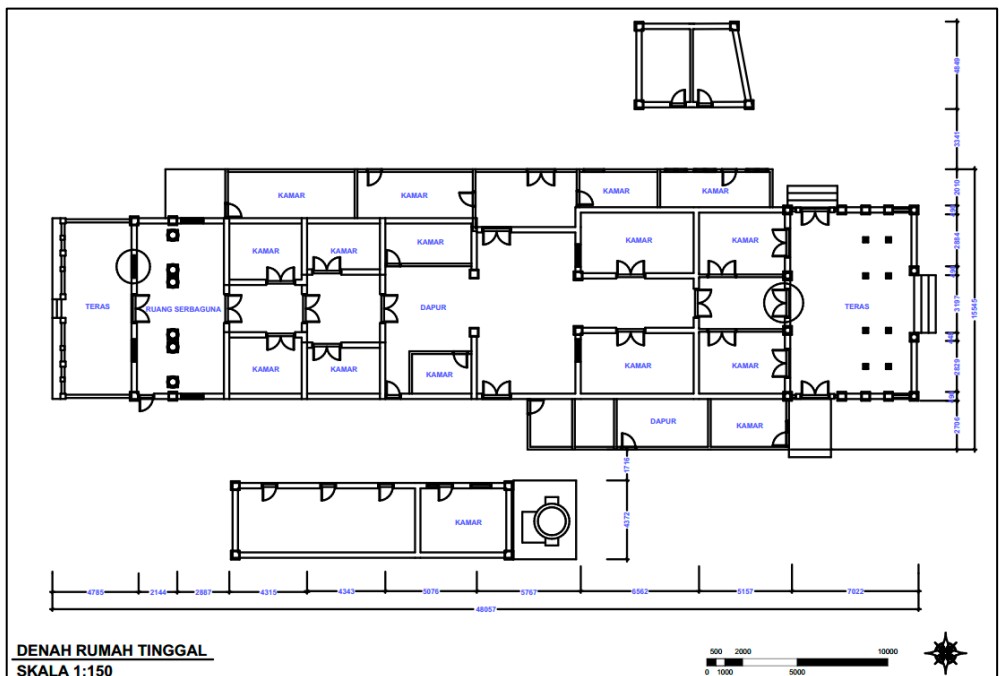

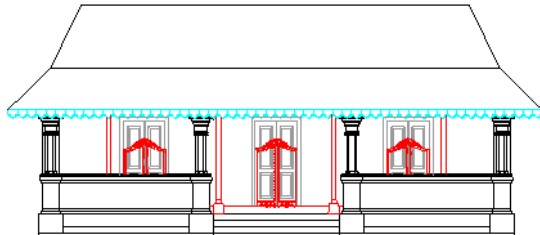

**Figure 2.** Architectural drawing (floor plan and elevation).

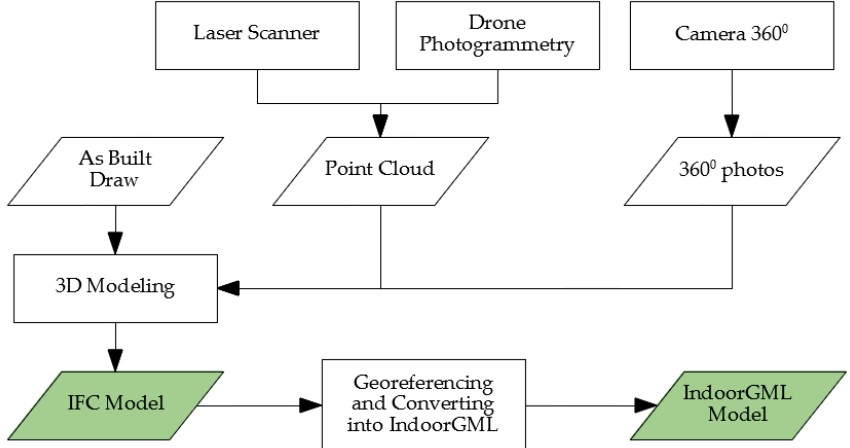

**Figure 3.** Flow diagram of 3D modeling.

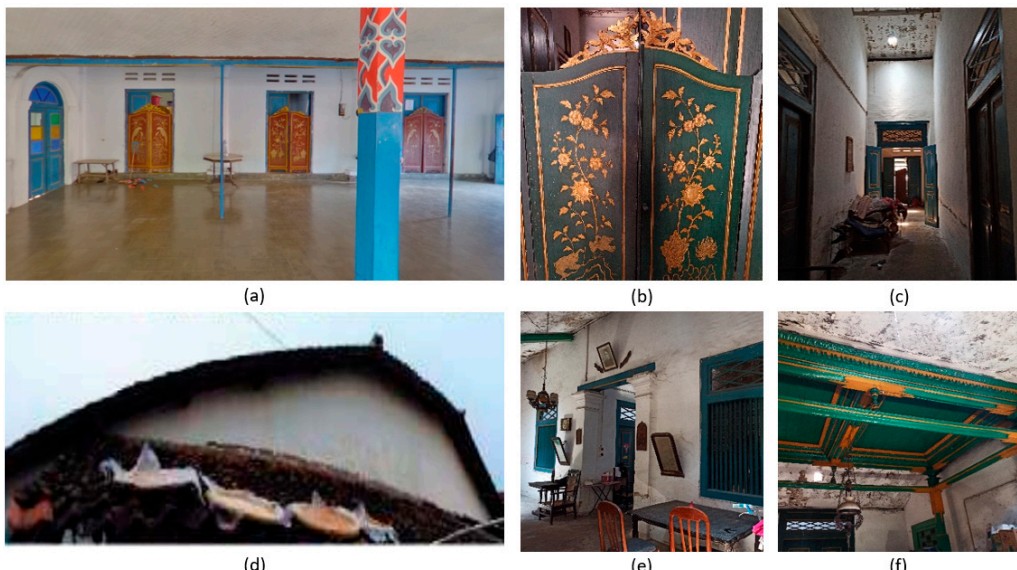

**Figure 4.** Building exterior and interior of Rumah Pangeran Arya Denda: (**a**) wooden column; (**b**) green painted door; (**c**) high roof and traditional doors; (**d**) traditional roof; (**e**) windows, large columns, and home decoration; (**f**) roof's ornament and lamp.

### 3.2. Building Assessment

Indonesia has a national standard document for buildings called Standar Nasional Indonesia (SNI). This document can be used as a reference for all buildings in this country, with detailed instructions and standards for construction and evaluation. In this research, the SNI 03-2396-2001 about natural room lighting and SNI 03-6572-2001 about room temperature were used for the building assessment. Various pieces of information from the building elements were needed to evaluate the buildings based on applicable regulations. However, in this study, the building element information was only the elements that appeared in the building model. The natural lighting evaluation workflow diagram can be seen in Figure 5, and the natural room temperature evaluation workflow can be seen in Figure 6.

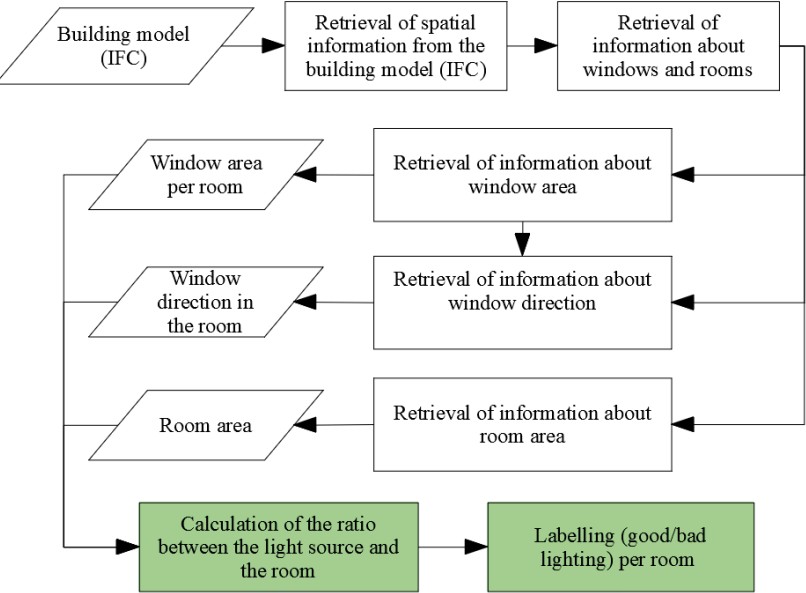

**Figure 5.** Workflow for natural lighting evaluation.

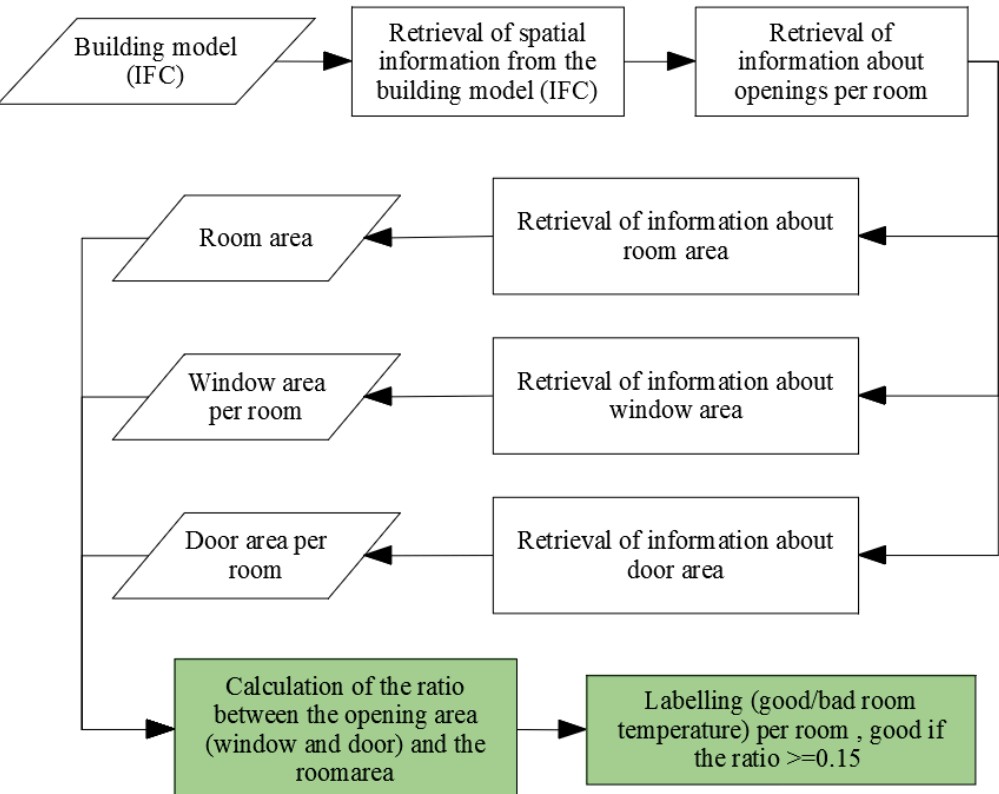

**Figure 6.** Workflow for room temperature evaluation.

The requirement information for natural lighting evaluation is the value of window area per room, window direction, and the value of room areas. The information needed for natural room temperature evaluation is the value of window area per room, door area per room, room area, building material, and room height. In this study, only information about the window, door, and room area was used for evaluation. The building materials and room heights were assumed to be reliable by applicable regulations.

The required building element information for building evaluation can be computed from the 3D building model (IFC) and compared with the standard. If the ratio between the building elements and the room meets the applicable standards, then the aspect of the building assessment is considered to meet the requirements. However, the overall building reliability assessment needs to be calculated again by considering other aspects of the assessment that were not calculated for this paper. An assessment of the reliability of the building considering other aspects will be carried out in a future article.

The analysis of indoor space and relation was carried out using the IndoorGML model. The information used was the size of the room, corridor, vertical access, and horizontal access. Vertical access can be in the form of stairs, escalators, or elevators, while horizontal access considers such items as doors and ramps. However, in the case study, only a one-story house was used; so, vertical access was not required. The accessibility analysis parameters we used referred to the Regulation of the Minister of Public Works and Public Housing No. 14/PRT/M/2017 concerning Requirements for Ease of Building Buildings and regulations from the United Kingdom, namely, The Building Regulations 2010: Fire Safety and The Building Regulations 2010: Access to and use of buildings. The natural lighting evaluation workflow diagram can be seen in Figure 7.

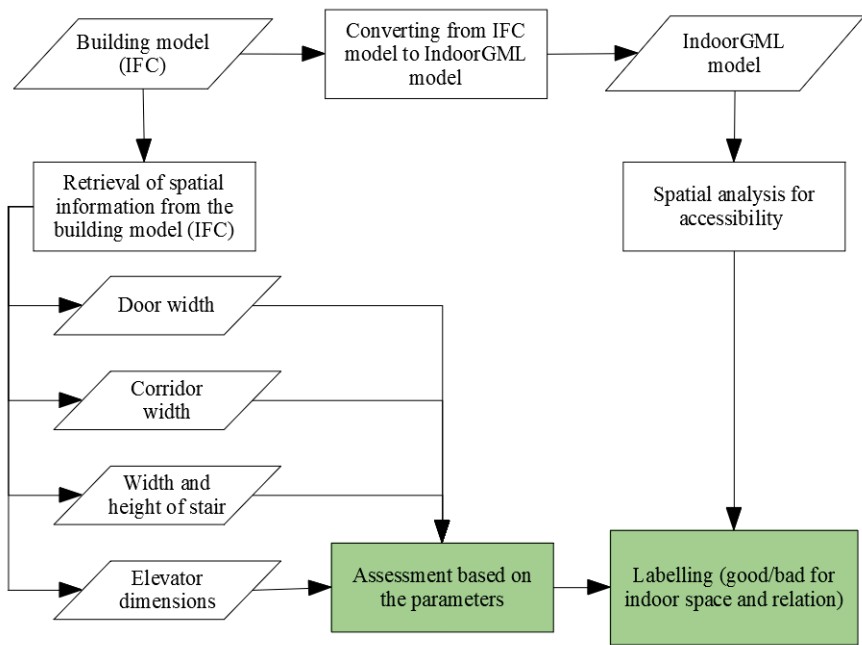

**Figure 7.** Workflow for indoor space and relation evaluation.

## 4. Results and Discussion

The results from the methods in the previous section can be divided into three subsections: the building as an IFC and IndoorGML model, the building evaluation, and a discussion of the results.

### 4.1. Building as IFC Model and IndorGML Model

Based on the handheld laser scanner data, the point cloud of the building's exterior and interior can be seen in Figure 8. The roof of the building was not visible from the terrestrial mapping; so, the digitized results of the roof shape or the shape of the roof in the 3D city model were used to complete the building model. A detailed 3D model can be generated using the combined laser scanner and photogrammetry data. The 3D building model in this study was created using the manual method based on the size measured from the point cloud. The result of the 3D building model can be seen in Figure 9. The building objects modeled were only exterior and interior in the form of rooms and openings. Meanwhile, architectural objects that were more detailed, such as ornaments or detailed forms of columns, were not modeled.

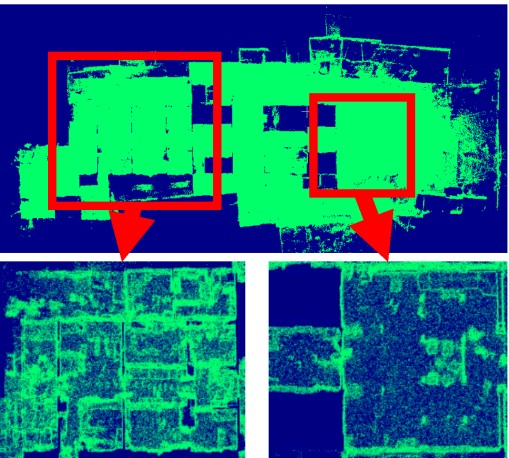

**Figure 8.** Point cloud from laser scanner for exterior and interior of Pangeran Arya's house.

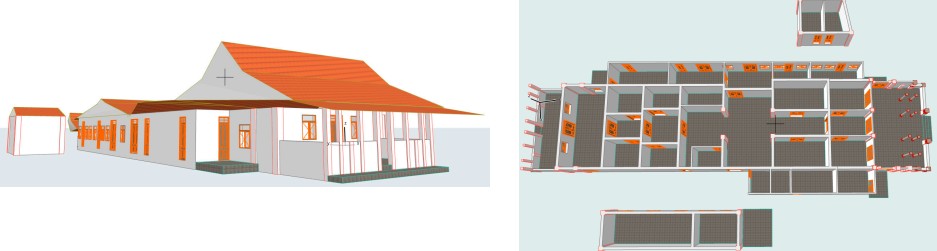

**Figure 9.** A 3D model for building exterior (**left**) and interior (**right**) based on combination data (laser scanner and photogrammetry).

The building model was a BIM model, where parts of the buildings were stored as properties in industry foundation classes (IFC). In this building model, there were several IFC properties, namely, IfcSpace for rooms, IfcWall for walls, IfcSlab for floors, IfcDoor for doors, IfcWindow for windows, IfcColumn for columns, and IfcBuildingElementProxy for roofs built as morph.

The IFC model only uses the local coordinates. Therefore, it must use georeferencing to make it sync with the GIS model; in this research, it was IndoorGML. Based on the diagram in Figure 3, georeferencing was carried out on the IFC model to have real coordinates. After the georeferencing process was complete, the conversion was carried out using the FME Workbench with the flow, as in [22], which was improved and adjusted to the needs. The IndoorGML model can be built into primal space and dual space models. Visualization of the IndoorGML's real space model can be seen in Figure 10. The dual space model was used to perform spatial analysis in the room, which will be formed into a graph, represented as nodes and lines. The green nodes represent the centroid of each room, and the red nodes represent the doors. The graph can be used to perform network analysis that will be described in the indoor spatial analysis part of this research.

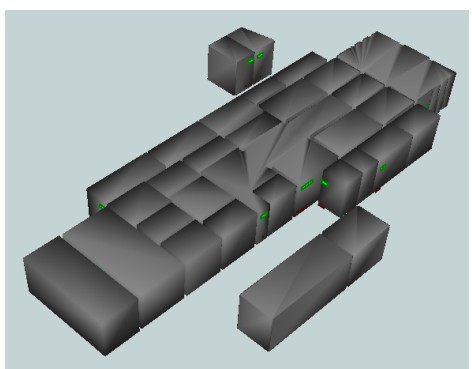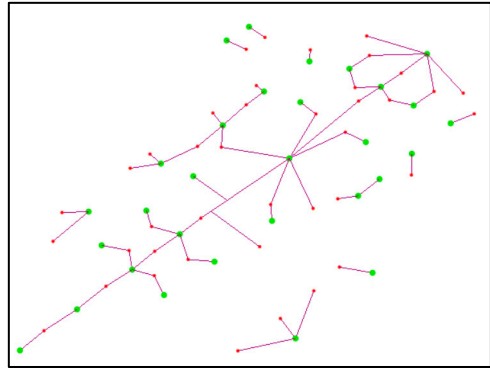

**Figure 10.** IndoorGML primal space (**left**) and dual space model (**right**) of Pangeran Arya's house.

### 4.2. Building Evaluation

Based on the 3D building model, the information of openings was taken and used for comparison with the room area. The information about the room was taken from IfcSpace, the window from IfcWindow, and the door from IfcDoor. Rooms' area and windows' area information was taken from IFC properties using IfcOpenShell, an open-source software library for working with the IFC file format from https://github.com/IfcOpenShell/IfcOpenShell (accessed 10 December 2021).

The length and width of the window were obtained using IfcOpenShell so that it could be accessed in the form of a table; several processes were carried out to obtain the window area and match the window ID with the room name. The floor plan with the name of the room from the model of the house of the Pangeran Arya building can be seen in Figure 11.

```python
import ifcopenshell
ifc_file = ifcopenshell.open('koica_ifc.ifc')
window = ifc_file.by_type('IfcProduct')
for product in window:
    print(product)
```

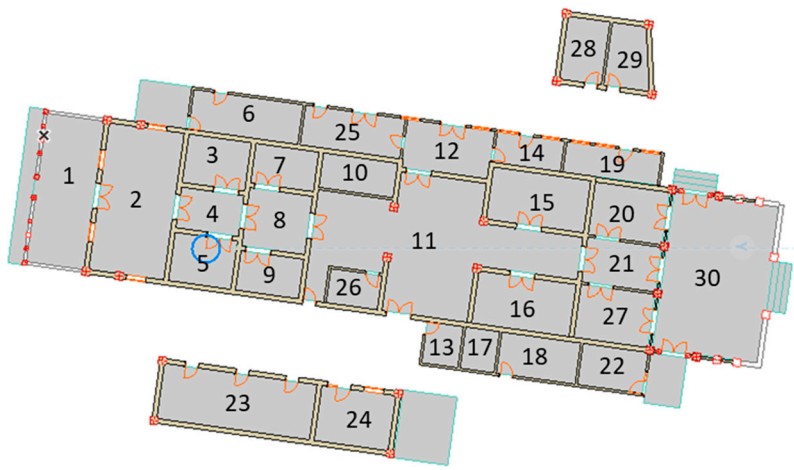

**Figure 11.** Building's plan with the zone number.

Based on the information taken from the IFC model, the ratio between the window area and the room area for natural lighting was calculated. As for natural ventilation, the ratio between the opening area and the room area was calculated. The results of the ratio calculations can be seen in the tables in Appendix A. In the national standard for reliable buildings, the ratio between window area and room area for natural lighting is 10%, while the ratio between opening area and room area for natural room ventilation is 15%. From Table A1, there were only five rooms that met the window requirements for natural lighting, which are shown by green cells. While almost all rooms met the minimum ratio for natural room ventilation, only room 16 did not meet the minimum ratio. In addition to ventilation and lighting systems, the indoor air condition also depends on the size of the room in the form of the height of the room. In this building, all rooms had a height above 2.8 m so that they met the minimum room height requirements.

The analysis of indoor space and relation was carried out using the IndoorGML model. The information used for this building only considered horizontal access, such as the doors' width and the distances from room to room. The doors' width information can be seen in Table A2. From the table, all doors met the minimum requirements for width, which was 1.2 m. The IndoorGML (dual space) model can be built into a node-relation graph to show the relationship or topology of the rooms in the building. By using a node-relation graph, it is possible to calculate the distance traveled from one room to another. In this study, an analysis was carried out for the distance from the room to the exit. There are several exits in this building, in the form of every existing exterior door. Based on the dual space of the IndoorGML model, the result of the distance from the room to the exterior door (exit) can be seen in Table A3. As stated in the table, all rooms had the distance to the nearest exit in accordance with the minimum requirements. The minimum required distance to the exit under The Building Regulations 2010: Fire safety is 45 m.

All results of the previous analysis were then used as input to calculate the reliability value of the building using the analytic hierarchy process (AHP) method [27]. There was a consensus from Puslitbang Permukiman (Settlement Research and Development Center) in 2011 that writes a building reliability assessment scheme by considering four reliability

criteria, namely, safety, health, comfort, and convenience. The weight percentage of each analysis can be seen in Figure 12. The percentage of safety criteria is not stated and only gives the description P, which means that these criteria are prerequisites that are considered reliable so that further evaluation of the reliability of the building can be carried out.

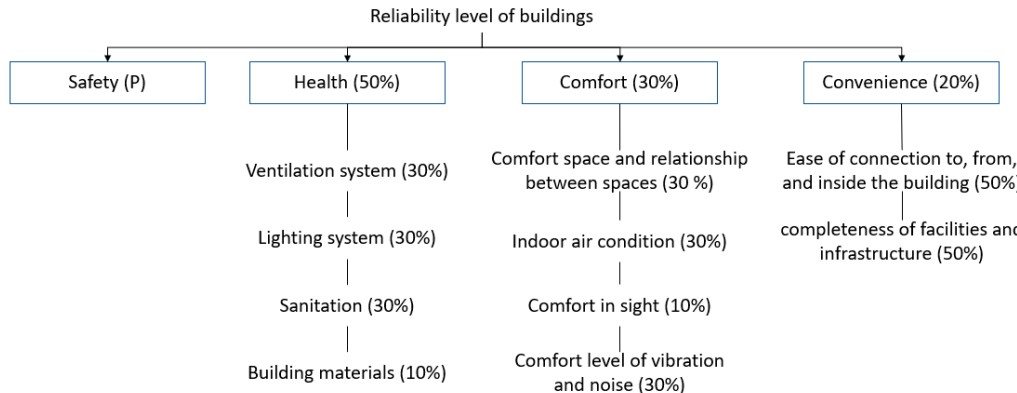

**Figure 12.** Building reliability assessment scheme based on consensus from Puslitbang Permukiman.

Based on the results of the analysis of several criteria, the evaluation tables for each criterion are shown in Table 1. In that table, Ss1 is for the ventilation system and Ss2 is for the lighting system, they are the parameters for Ss (health). Sn1 is for comfort space and Sn2 is for indoor air condition, they are the parameters for Sn (comfort). Sm1 is for the accessibility (ease of connection), the parameter for Sm (convenience). The calculation value scale in the table is from the numbers 1–5, with a score from 20 to 100. A building is said to be reliable if it meets two requirements: (1) meets the prerequisites P and (2) scores Ss (Health), Sn (Comfort), and Sm (Convenience) are greater than 60. The final score of a reliable building is achieved when the score is Sk ≥ 60. Based on the information obtained and the criteria for assessing the reliability of the building, only a few criteria were selected that could be met based on the available information.

**Table 1.** Score for each criterion.

| Amount/Value | Ss1 | Ss2 | Sn1 | Sn2 | Sm1 |
|---|---|---|---|---|---|
| Parameter | 1 | 1 | 1 | 1 | 1 |
| Total Sample | 28 | 28 | 40 | 28 | 30 |
| Sample Complied | 27 | 5 | 40 | 28 | 30 |
| Value | 5 | 2 | 5 | 5 | 5 |
| Score | 100 | 40 | 100 | 100 | 100 |

By using the relative weights of the AHP analysis, a recapitulation of the score calculation was obtained (Table 2). From the table, it is found that the initial score was Ss = 70, Sn = 100, and Sm = 100 so that all of them met the minimum score. Then, the total score of building reliability after calculation was Sk = 85; so, it was categorized as a reliable building.

### 4.3. Discussion

Currently, Pangeran Arya's house is divided into two parts by two of his descendants who inhabit the house. There are several additional building elements from the original building due to the needs of the families who inhabit them, but the original shapes and ornaments are still maintained. There are ornaments on doors, windows, and columns that cannot be modeled using the collected data as a point cloud. For example, the point cloud result of the room can be seen in Figure 13 as Zone 30 (the photo image of this room

can be seen in Figure 4a), where the point cloud only shows the shape of columns and doors without any ornaments. In other words, if the point cloud data are combined with the photo image and a more detailed point cloud, using a handheld laser scanner for each building element, it is possible to model the building interior with more detail including the ornaments.

**Table 2.** Result for each criterion (weighting method).

| No. | Criteria/Sub-Criteria | Initial Score | Weight (%) | Final Score |
|---|---|---|---|---|
| 1 | **Health (Ss)** | **70** | **50** | **35** |
| 2 | Ventilation System (Ss1) | 100 | | |
| 3 | Lighting System (Ss2) | 40 | | |
| 4 | **Comfort (Sn)** | **100** | **30** | **30** |
| 5 | Comfort Space (Sn1) | 100 | | |
| 6 | Indoor Air Condition (Sn2) | 100 | | |
| 7 | **Convenience (Sm)** | **100** | **20** | **20** |
| 8 | Accessibility (Sm1) | 100 | | |
| | **Final Score (Sk)** | | | **85** |

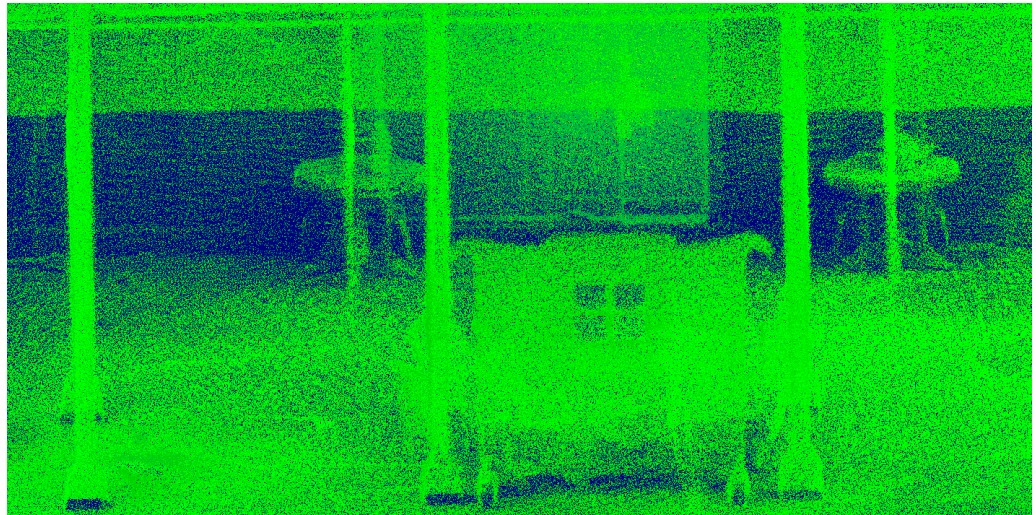

**Figure 13.** Point cloud of zone 30 from laser scanner.

The building model can be used for some building assessment or evaluation; the evaluation aspects are natural lighting, natural room ventilation, and indoor space and relation. Standards and regulations used for this assessment were for a building as a residence, while those used as case studies in this research were heritage buildings. Therefore, when an inappropriate evaluation is obtained, the building is not demolished but conservation is carried out. This building model then can be used for the documentation of heritage buildings for further conservation. However, for the results of evaluating the reliability of the Pangeran Arya's house, it was found that the building was considered reliable. The results are considered reliable because there are several criteria that were calculated; although, in one criterion in the form of natural lighting, the results were not good. These results only took into consideration a few criteria based on the information found so that more complete information can be collected and a more complete evaluation of the reliability of the building can also be carried out.

The room ventilation and room lighting evaluation can be used for other building functions with parameters based on different regulations. For additional criteria, information for lighting and room temperature can be added in the form of lighting and mechanical ventilation so that the actual room conditions can be represented properly. For the evaluation of the reliability of other buildings, we can take information from the IFC model similar to this but with some modifications according to information needs. This evaluation can be used by the technical examiner of the reliability of the building without the need to come directly to the site, only through the building model, if the model has enough information for assessment.

## 5. Conclusions

A combination of acquisition methods, photogrammetry, and laser scanning technology can be performed to model the building. To generate the building model, we can use a combination of aerial and terrestrial mapping. The existing buildings were modeled in LOD 3, in which the building exterior and interior were modeled with more detail, and the building model information can be used for building evaluation by spatial analysis based on the rules and regulations in Indonesia. Based on the created model, a building assessment in this research was performed by comparing 3D models of buildings that have been built with existing regulations/standards for room temperature, natural lighting, and indoor space and relation evaluation.

Currently, this heritage building is used as a residence, and the regulation used for this assessment was for a building as a residence. From the result of all the rooms in the building (Pangeran Arya Denda's house), the result of evaluating a building was reliable; although, in one criterion, in the form of natural lighting, the results were not good. However, the building that was used as a case study is a heritage building that is used as a vernacular object and requires conservation; so, this model is more likely to be used for evaluation before documentation and conservation.

The building reliability assessment method in this preliminary study can be complemented by the addition of various other pieces of information on the building model, so that the assessment will be better and in accordance with the needs of technical examiners. For a complete documentation of the heritage building, ornaments on the building parts are needed. For further research, a handheld laser scanner will be used so that each ornament can be modeled in more detail. This preliminary study can be used as a starting point for evaluating building reliability with more complete building reliability criteria as well as the documentation of heritage buildings for conservation purposes.

**Author Contributions:** Conceptualization, S.W.T., D.S., K.W. and I.P.; methodology, S.W.T. and D.S.; software, D.S.; validation, S.W.T. and D.S.; formal analysis, S.W.T. and D.S.; investigation, S.W.T. and D.S.; resources, D.S.; data curation, D.S. and I.P.; writing—original draft preparation, S.W.T.; writing—review and editing, D.S., I.P. and K.W.; visualization, S.W.T. and D.S.; supervision, D.S., I.P. and K.W.; project administration, S.W.T. and D.S.; funding acquisition, D.S. All authors have read and agreed to the published version of the manuscript.

**Funding:** This research received funding from the Lembaga Pengelola Dana Pendidikan (LPDP No. 201909210215479), Indonesia.

**Data Availability Statement:** The data presented in this study are available on request from the corresponding author. The data are not publicly available because it is related to the privacy of the building owner.

**Conflicts of Interest:** The authors declare no conflict of interest.

## Appendix A

**Table A1.** Room area and windows area for building's rooms in house of Pangeran Arya. The green cell indicates that the value meets the minimum requirements in the regulations, while the red cell indicates that the value does not meet the minimum requirements.

| Room | Room Area (m²) | Windows' Area (m²) | Door Area (m²) | Ratio W/R | Ratio (W + D)/R |
|---|---|---|---|---|---|
| Zone 2 | 45.83 | 7.83 | 9.6 | 0.17 | 0.38 |
| Zone 3 | 12.49 | 0.00 | 4.8 | 0.00 | 0.38 |
| Zone 4 | 11.39 | 0.00 | 19.2 | 0.00 | 1.69 |
| Zone 5 | 12.49 | 0.00 | 4.8 | 0.00 | 0.38 |
| Zone 6 | 18.00 | 0.00 | 5.1 | 0.00 | 0.28 |
| Zone 7 | 10.27 | 0.00 | 4.8 | 0.00 | 0.47 |
| Zone 8 | 16.07 | 0.00 | 19.2 | 0.00 | 1.19 |
| Zone 9 | 10.27 | 0.00 | 4.8 | 0.00 | 0.47 |
| Zone 10 | 10.60 | 0.00 | 2.55 | 0.00 | 0.24 |
| Zone 11 | 93.77 | 0.00 | 33.9 | 0.00 | 0.36 |
| Zone 12 | 17.34 | 3.23 | 14.7 | 0.19 | 1.03 |
| Zone 13 | 6.16 | 0.00 | 2.55 | 0.00 | 0.41 |
| Zone 14 | 8.47 | 5.79 | 5.25 | 0.68 | 1.30 |
| Zone 15 | 21.01 | 0.00 | 4.8 | 0.00 | 0.23 |
| Zone 16 | 93.47 | 0.00 | 4.8 | 0.00 | 0.05 |
| Zone 17 | 5.76 | 0.00 | 2.55 | 0.00 | 0.44 |
| Zone 18 | 13.43 | 0.00 | 2.25 | 0.00 | 0.17 |
| Zone 19 | 11.90 | 5.87 | 2.7 | 0.49 | 0.72 |
| Zone 20 | 16.44 | 0.00 | 9.6 | 0.00 | 0.58 |
| Zone 21 | 13.92 | 0.00 | 19.2 | 0.00 | 1.38 |
| Zone 22 | 11.10 | 0.00 | 4.5 | 0.00 | 0.41 |
| Zone 23 | 34.89 | 0.00 | 7.65 | 0.00 | 0.22 |
| Zone 24 | 18.92 | 1.96 | 4.5 | 0.10 | 0.34 |
| Zone 25 | 16.40 | 0.45 | 9.9 | 0.03 | 0.63 |
| Zone 26 | 7.24 | 0.00 | 2.55 | 0.00 | 0.35 |
| Zone 27 | 18.18 | 0.00 | 9.6 | 0.00 | 0.53 |
| Zone 28 | 10.85 | 0.00 | 2.55 | 0.00 | 0.24 |
| Zone 29 | 10.34 | 0.00 | 2.55 | 0.00 | 0.25 |

**Table A2.** Door width and height in house of Pangeran Arya. The green cell indicates that the value meets the minimum requirements in the regulations.

| Door ID | Width | Height | Door ID | Width | Height |
|---|---|---|---|---|---|
| DOO-019 | 1.60 m | 3.00 m | DOO-023 | 0.85 m | 3.00 m |
| DOO-015 | 1.60 m | 3.00 m | DOO-037 | 0.90 m | 3.00 m |
| DOO-014 | 1.60 m | 3.00 m | DOO-022 | 0.85 m | 3.00 m |
| DOO-016 | 1.60 m | 3.00 m | DOO-037 | 0.90 m | 3.00 m |
| DOO-017 | 1.60 m | 3.00 m | DOO-005 | 1.60 m | 3.00 m |
| DOO-026 | 0.85 m | 3.00 m | DOO-006 | 1.60 m | 3.00 m |
| DOO-027 | 0.85 m | 3.00 m | DOO-001 | 1.60 m | 3.00 m |
| DOO-018 | 1.60 m | 3.00 m | DOO-002 | 1.60 m | 3.00 m |
| DOO-012 | 1.60 m | 3.00 m | DOO-036 | 1.50 m | 3.00 m |
| DOO-013 | 1.60 m | 3.00 m | DOO-030 | 0.85 m | 3.00 m |
| DOO-004 | 1.60 m | 3.00 m | DOO-031 | 0.85 m | 3.00 m |
| DOO-007 | 1.60 m | 3.00 m | DOO-032 | 0.85 m | 3.00 m |
| DOO-008 | 1.60 m | 3.00 m | DOO-035 | 1.50 m | 3.00 m |
| DOO-009 | 1.60 m | 3.00 m | DOO-027 | 0.85 m | 3.00 m |
| DOO-011 | 1.60 m | 3.00 m | DOO-028 | 0.85 m | 3.00 m |
| DOO-024 | 0.85 m | 3.00 m | DOO-003 | 1.60 m | 3.00 m |
| DOO-025 | 0.85 m | 3.00 m | DOO-033 | 0.85 m | 3.00 m |

**Table A2.** *Cont.*

| Door ID | Width | Height | Door ID | Width | Height |
|---------|-------|--------|---------|-------|--------|
| DOO-010 | 1.60 m | 3.00 m | DOO-034 | 0.85 m | 3.00 m |
| DOO-028 | 0.85 m | 3.00 m | DOO-003 | 1.60 m | 3.00 m |
| DOO-029 | 0.85 m | 3.00 m | DOO-021 | 1.60 m | 2.13 m |

**Table A3.** Distance of each room to the nearest exterior door. The green cell indicates that the value meets the minimum requirements in the regulations.

| Room | Distance (m) | Room | Distance (m) | Room | Distance (m) |
|------|--------------|------|--------------|------|--------------|
| Zone 1 | 2 | Zone 11 | 5.4 | Zone 21 | 7.8 |
| Zone 2 | 6.6 | Zone 12 | 1.7 | Zone 22 | 2.3 |
| Zone 3 | 15.0 | Zone 13 | 1.5 | Zone 23 | 1.9 |
| Zone 4 | 11.3 | Zone 14 | 1.0 | Zone 24 | 2.2 |
| Zone 5 | 14.7 | Zone 15 | 12.7 | Zone 25 | 1.4 |
| Zone 6 | 1.9 | Zone 16 | 12.5 | Zone 26 | 12.0 |
| Zone 7 | 10.7 | Zone 17 | 3.9 | Zone 27 | 4.8 |
| Zone 8 | 7.0 | Zone 18 | 2.3 | Zone 28 | 2.0 |
| Zone 9 | 11.2 | Zone 19 | 1.5 | Zone 29 | 2.0 |
| Zone 10 | 8.1 | Zone 20 | 5.1 | Zone 30 | 3.5 |

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
