# Peer review of "A Preliminary Study of 3D Vernacular Documentation for Conservation and Evaluation: A Case Study in Keraton Kasepuhan Cirebon"

_buildings, doi:10.3390/buildings13020546_

Round 1

Reviewer 1 Report

In the introduction, the authors clearly present the type of research carried out on the building under study and its purposes, also offering a pertinent overview of the different technologies used for the survey and 3D modeling of buildings and in particular those of historical interest .

The cited references appear punctual and pertinent to the research.

The description of the methodologies, instruments and software used in the three-dimensional representation of the building is well structured, however in some cases the figures could be improved. For example fig. 2 is not able to return the scale reference, even if you enlarge the image the measurements are not readable and the absence of a metric scale does not help.

The conclusions adequately summarize the results achieved and open the way for a more detailed study of the good.

It is therefore advisable to publish after a minimum revision by the authors.

Author Response

Dear reviewer,

Improvements have been made to figure 2 by adding a scale bar.

Thanks for all the comments and suggestions. The improvements made can clarify the contents of this paper.

Reviewer 2 Report

Paper written clearly and completely. Of considerable importance for the world of BIM

Author Response

Dear reviewer,

Thank you for the review and i have made some improvement on the paper.

Reviewer 3 Report

This paper protects and evaluates the cultural heritage by BIM and IndoorGML. In general, it is a good and interesting study. But there are still some problems in the paper.

1. In the abstract, the research core of the paper does not clearly been described, especially the method of building evaluation.

2. In 3.2 building evaluation, I have some doubts about the method of natural lighting evaluation. It is arbitrary to evaluate natural lighting only from the parameters, such as window area, direction, etc. How to consider the external environment, shelter and other factors of the building. Please explain clearly.

3. The conclusion is not clear. What is the research conclusion, what is the innovation, what is the follow-up research.

4. Some language expressions are not strongly related to the theme of the paper, so it is suggested to modify them. For example: handheld laser scanning (Line 244, Line 401), etc.

5. Although the theme of the paper emphasizes the evaluation of the protection of cultural heritage, it does not clearly express the particularity of cultural heritage buildings, and the difference from the evaluation of ordinary buildings.

Author Response

Dear reviewer,

Here are the responses to your comments.

  1. Abstract has been improved by adding a method for building evaluation
  2. In the regulations regarding the calculation of lighting in buildings, various parameters are used. As with the other buildings analysis, a variety of information is needed. However, in this study, only building elements that appear in the building model are used as parameters in the analysis. So that the assumption has been added in the paper that the parameters used in this study are only elements that appear in the building model to clarify the analytical method in this study.
  3. The conclusion has been improved.
  4. A handheld laser scanner was introduced in this study as one of the tools used for data acquisition. In this study, the data acquisition method used in building modeling is not clearly written because it has been explained in the previous paper in reference [23]. Then, a handheld laser scanner appears in conclusion (line 401) as a form of a plan for further research to document heritage building elements in more detail at longer distances, close to each of these elements.
  5. The peculiarities of the cultural heritage buildings in this study are shown in section 3.1, line 180. Meanwhile, there are no differences in evaluating cultural heritage buildings and modern ones because, currently, the regulations for assessing cultural and modern buildings are the same. However, what can be different is the evaluation of building reliability based on building functions, such as office, residential, and hotel functions.

Thanks for all the comments and suggestions. The improvements made can clarify the contents of this paper.